# Mask family proteins ANKHD1 and ANKRD17 regulate YAP nuclear import and stability

Clara Sidor[1], Nerea Borreguero-Munoz[1], Georgina C Fletcher[1], Ahmed Elbediwy[1], Oriane Guillermin[1], Barry J Thompson[1,2]*

[1]Epithelial Biology Laboratory, Francis Crick Institute, London, United Kingdom; [2]EMBL Australia, ACRF Department of Cancer Biology and Therapeutics, John Curtin School of Medical Research, The Australian National University, Canberra, Australia

**Abstract** Mask family proteins were discovered in *Drosophila* to promote the activity of the transcriptional coactivator Yorkie (Yki), the sole fly homolog of mammalian YAP (YAP1) and TAZ (WWTR1). The molecular function of Mask, or its mammalian homologs Mask1 (ANKHD1) and Mask2 (ANKRD17), remains unclear. Mask family proteins contain two ankyrin repeat domains that bind Yki/YAP as well as a conserved nuclear localisation sequence (NLS) and nuclear export sequence (NES), suggesting a role in nucleo-cytoplasmic transport. Here we show that Mask acts to promote nuclear import of Yki, and that addition of an ectopic NLS to Yki is sufficient to bypass the requirement for Mask in Yki-driven tissue growth. Mammalian Mask1/2 proteins also promote nuclear import of YAP, as well as stabilising YAP and driving formation of liquid droplets. Mask1/2 and YAP normally colocalise in a granular fashion in both nucleus and cytoplasm, and are co-regulated during mechanotransduction.

*For correspondence:
barry.thompson@crick.ac.uk

Competing interests: The authors declare that no competing interests exist.

## Introduction

The YAP/TAZ family of oncoproteins has a single homolog in *Drosophila* named Yorkie (Yki) that was discovered to control tissue growth in proliferating epithelia (*Huang et al., 2005*). Genetic analysis of YAP and TAZ in mice is revealing an important role for both proteins in driving cell proliferation during tissue regeneration as well as during formation of several tumour types (*Cai et al., 2015*; *Cai et al., 2010*; *Camargo et al., 2007*; *Chen et al., 2014*; *Dong et al., 2007*; *Elbediwy et al., 2016*; *Gruber et al., 2016*; *Reginensi et al., 2015*; *Schlegelmilch et al., 2011*; *Vincent-Mistiaen et al., 2018*; *Zhang et al., 2011*). Yki/YAP/TAZ were shown to function as transcriptional co-activators of the nuclear DNA binding transcription factors TEAD1-4 (named Scalloped in *Drosophila*) (*Koontz et al., 2013*; *Wu et al., 2008*). The molecular mechanisms by which Yki/YAP/TAZ are physiologically regulated are still being determined.

One key mechanism regulating Yki/YAP/TAZ subcellular localisation is phosphorylation by the Hippo pathway kinase Warts/LATS on five Serine residues, which induces retention in the cytoplasm by binding to 14-3-3 proteins (*Huang et al., 2005*; *Oh and Irvine, 2008*; *Oh and Irvine, 2009*; *Zhao et al., 2007*). In response to mechanical stretching/flattening of cells, both Yki (*Fletcher et al., 2018*) and YAP/TAZ (*Dupont et al., 2011*; *Wada et al., 2011*; *Zhao et al., 2007*) can translocate from the cytoplasm to the nucleus. Both Yki (*Manning et al., 2018*) and YAP (*Ege et al., 2018*) undergo dynamic and continuous nuclear-cytoplasmic shuttling which must involve specific nuclear import and export factors. Several regulators of YAP nuclear export have been proposed (*Furukawa et al., 2017*; *Lee et al., 2018*). In contrast, no nuclear import factors for the Yki/YAP/TAZ

family have been identified and these proteins lack a conventional nuclear localisation sequence (NLS) (*Kofler et al., 2018*; *Wang et al., 2016*).

Here we show that the Mask family of ankyrin-repeat domain proteins, which feature conserved NLS and NES (nuclear export sequence) motifs, mediate nuclear import of both *Drosophila* Yki and mammalian YAP. Previous work identified an essential requirement for *Drosophila* Mask and its mammalian homologs Mask1 (ANKHD1) and Mask2 (ANKRD17) in promoting Yki/YAP transcriptional activity, but the mechanism by which Mask family proteins act has remained unclear (*Dong et al., 2016*; *Machado-Neto et al., 2014*; *Sansores-Garcia et al., 2013*; *Sidor et al., 2013*). We find that loss of Mask family proteins prevents nuclear import of Yki/YAP in both mammalian cells and *Drosophila*. Furthermore, while Mask is normally required for Yorkie to drive tissue growth, addition of an ectopic NLS to Yki is sufficient to bypass this requirement in *Drosophila*. Double conditional knockout of *Mask1/ANKHD1* and *Mask2/ANKRD17* in mouse intestinal organoids, together with siRNA knockdown of these proteins in human intestinal cells, confirms an essential requirement for Mask proteins in YAP nuclear import and stability. Finally, we show that overexpression of Mask1/2 is sufficient to stabilise YAP protein levels and can also drive phase separation of YAP into liquid droplets, suggesting that colloidal phase separation may contribute to the regulation of YAP activity.

## Results

We began by examining whether Mask family proteins have a role in regulating the subcellular localisation of Yki, as we were unable to identify a direct transcriptional activation function for Mask in a GAL4 reporter assay (*Figure 1—figure supplement 1*). Previously, we ruled out a possible role for Mask in promoting Yki nuclear import based on antibody staining for Yki in $mask^{10.22}$ null mutant clones in the *Drosophila* wing disc, where Yki is mostly cytoplasmic (*Sidor et al., 2013*). Recently, a Yki-GFP knock-in line revealed robust nuclear localisation of Yki in the mechanically stretched cells of the *Drosophila* ovarian follicle cell epithelium (*Fletcher et al., 2018*). We therefore induced $mask^{10.22}$ null mutant clones induced in the developing follicle cell epithelium, in which an endogenously tagged Yki-GFP knock-in is cytoplasmic at stage 10 but becomes strongly nuclear during stage 11 as the columnar cells are stretched mechanically (*Fletcher et al., 2018*) (*Figure 1A,B*). We find that Yki-GFP is lost from the nucleus and accumulates in the cytoplasm in *mask* mutant cells (*Figure 1C–F*). These findings indicate that Mask proteins are required for normal nuclear localisation of Yki.

The above observations suggest a potential role for Mask in nuclear import of Yki. Mask family proteins have a conserved nuclear localisation signal (NLS) located just C-terminal to the second ankyrin repeat domain (*Figure 2A*). To test whether this NLS is required for the function of Mask in vivo, we generated a *mask* CRISPR-knockin lacking the NLS motif (*maskΔNLS)*, which was homozygous lethal but showed normal expression of the mutant protein and localisation to the cytoplasm in clones (*Figure 2—figure supplement 1*). Consistent with a role for Mask in nuclear import, clones of follicle cells homozygous for the *maskΔNLS* allele show a strong decrease in nuclear Yki-GFP and a corresponding increase in the level of cytoplasmic Yki-GFP (*Figure 2B–D*). These results demonstrate the essential requirement for the NLS motif in Mask function and support a key role for Mask in Yki nuclear import.

Since the phenotypic characterisation of *mask* mutants focused on the proliferating epithelia of *Drosophila*, such as the developing wing (*Sansores-Garcia et al., 2013*; *Sidor et al., 2013*), we sought to examine the role of Mask in regulating Yki-GFP in this tissue. Yki is known to be primarily cytoplasmic in the developing wing, which is composed of densely packed columnar epithelial cells (*Oh and Irvine, 2008*) (*Figure 3A*). Since clones overexpressing Yki (MARCM clones expressing *tub.Gal4 UAS.Yki*) were shown to require Mask in order to drive cell proliferation in the wing (*Sidor et al., 2013*), we examined whether Mask affects the nuclear localisation of overexpressed Yki. We find that overexpressed Yki is readily detected in both nucleus and cytoplasm in control MARCM clones, but not in *mask* mutant MARCM clones, where the level of nuclear Yki is reduced relative to the level of Yki in the cytoplasm (*Figure 3A*). To test the function of Mask in nuclear import of endogenous Yki, we examined the peripodial epithelium, which features strongly nuclear Yki-GFP. We find that silencing of Mask expression by RNAi in the peripodial epithelium with *Ubx.Gal4*, prevents the normally strongly nuclear Yki-GFP localisation in these cells (*Figure 3B*). These

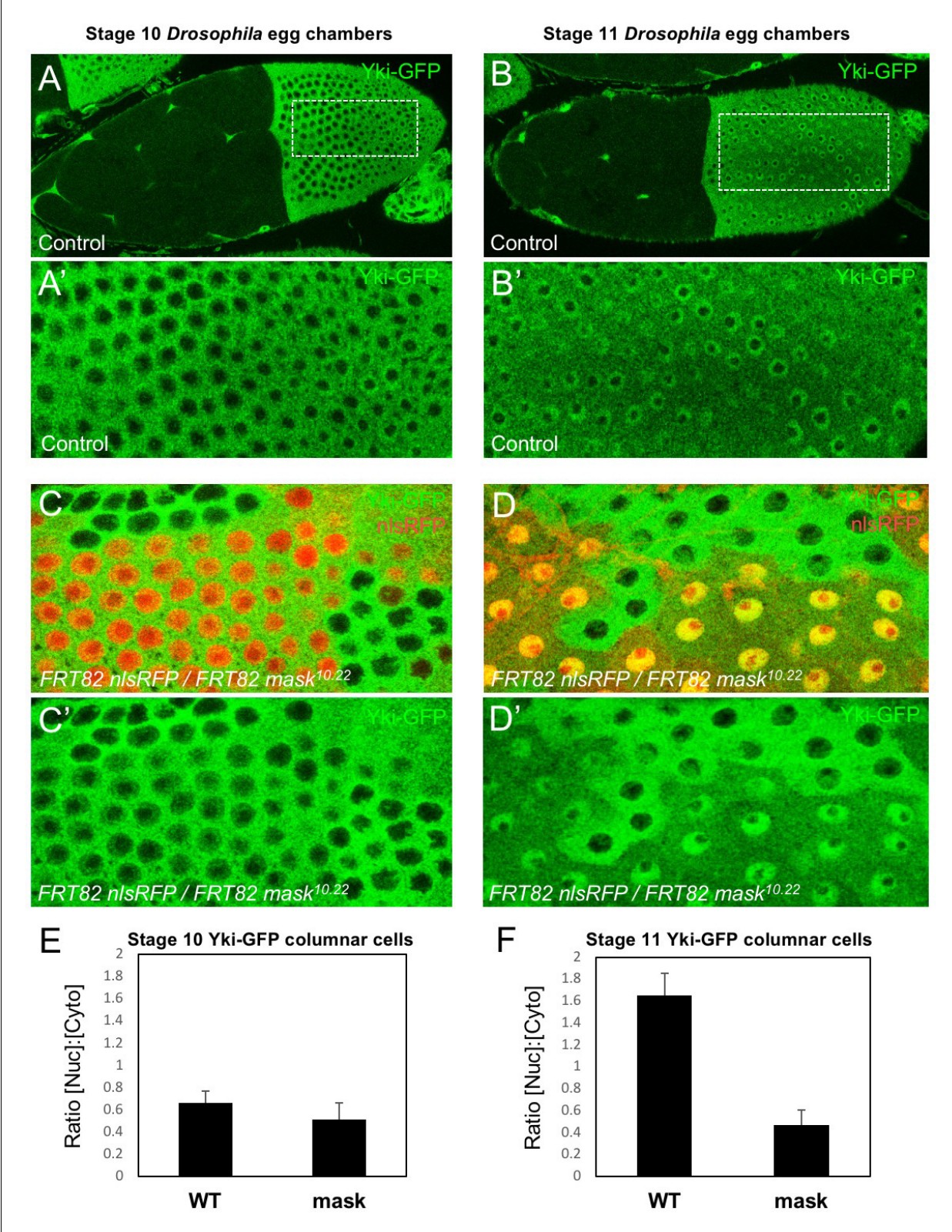

**Figure 1.** Mask is required to promote nuclear localisation of Yki in *Drosophila* follicle cells. (**A**) Stage 10 *Drosophila* egg chamber with endogenously tagged Yki-GFP (green) localised to the nucleus of stretch cells (anterior) and cytoplasm of columnar cells (posterior). (**A'**) Magnification of columnar cells. (**B**) Stage 11 *Drosophila* egg chamber with endogenously tagged Yki-GFP (green) localised to the nucleus of stretch cells (anterior) and nucleus of flattening columnar cells caused by growth of the oocyte (posterior). (**B'**) Magnification of flattening columnar cells. (**C**) Stage 10 *Drosophila* egg

*Figure 1 continued on next page*

*Figure 1 continued*

chamber containing null mutant clones of *mask* (marked by absence of nuclear RFP, red) display cytoplasmic Yki-GFP. (**C'**) Yki-GFP single channel. (**D**) Stage 11 *Drosophila* egg chamber containing null mutant clones of *mask* (marked by absence of nuclear RFP, red) display cytoplasmic Yki-GFP. (**D'**) Yki-GFP single channel. (**E**) Quantification of nuclear:cytoplasmic ratio of Yki-GFP in (**C**) n = 7 clones. (**F**) Quantification of nuclear:cytoplasmic ratio of Yki-GFP in (**D**) n = 12 clones.

The online version of this article includes the following figure supplement(s) for figure 1:

**Figure supplement 1.** Mask has no intrinsic transcriptional co-activator activity in a GAL4-UAS reporter assay.

results confirm that Mask is required for the nuclear localisation of both endogenous and overexpressed Yki in the developing wing of *Drosophila*.

The above results suggest that the reduced level of nuclear Yki may account for the failure of overexpressed Yki to drive cell proliferation in *mask* mutant clones in the developing wing. To test this notion, we linked an ectopic nuclear localisation sequence (NLS) and an epitope tag (HA) to Yki and expressed it in wild-type and *mask* mutant MARCM clones in the wing. We confirm that MARCM clones expressing *tub.Gal4 UAS.Yki-NLS-HA* are able to restore both nuclear localisation of Yki and cell proliferation in *mask* mutant cells (*Figure 4A–G*; *Sidor et al., 2013*). We note that even Yki-NLS-HA is still mildly less nuclear and more cytoplasmic when expressed in *mask* mutant cells compared to wild-type cells, and that this may therefore explain the mild difference in growth between these two types of clones (*Figure 4E–G*). We find similar results in the eye imaginal disc (*Figure 4H–K*). Thus, linkage of a nuclear localisation sequence to Yki can largely bypass the requirement for *mask* in Yki-driven cell proliferation in vivo. These results indicate that the primary function of Mask is to promote nuclear localisation of Yki in *Drosophila*.

To extend these findings to mammals, we performed siRNA knockdown experiments in human HEK293T and Caco2 cells. In both cases, silencing of Mask1 (ANKHD1) was sufficient to reduce both nuclear localisation of YAP and total YAP levels, as measured by immunostaining or western blotting of cell lysates (*Figure 5A–C*). Notably, double siRNA against Mask1/2 causes apoptosis of transfected cells, indicating that the two proteins act redundantly (*Figure 5C*; *Figure 5—figure supplement 1*). Furthermore, we generated double conditional floxed mice for Mask1 (ANKHD1) and Mask2 (ANKRD17). The double floxed Mask1/2 mice were used to generate intestinal organoids, which were then infected with GFP-tagged Adenoviral Cre (AdCre-GFP) to drive deletion of Mask1/2 in clones. Clonal deletion of Mask1/2 resulted in a strong reduction in YAP levels compared to surrounding wild-type cells (*Figure 5D*) and clone sizes were typically only 1–2 cells, with cells frequently undergoing extrusion from the epithelium and death (*Figure 5—figure supplement 2*). We find a similar effect of Mask1 siRNA on TAZ localisation (*Figure 5—figure supplement 3*). These findings show that mammalian Mask1/2 have a conserved function in nuclear import of YAP but are additionally required to stabilise the YAP protein.

We next tested whether overexpression of Mask1/2 might be sufficient to cause mis-regulation of YAP in human cells. We find that overexpression of either Mask1 or Mask2 caused strong stabilisation of YAP protein levels in Caco2 cells (*Figure 6A,B*). Furthermore, overexpression of Mask1 in HEK293T cells, which are known to generate high levels of expression, caused formation of abnormally large droplets containing both Mask1 and YAP proteins (*Figure 6*). We speculate that Mask proteins may cluster/polymerise YAP to drive colloidal phase separation – a classic mechanism of cellular compartmentalisation (*Hardy, 1899*; *Iborra, 2007*; *Walter and Brooks, 1995*; *Wilson, 1899*). Recent work has shown that cells expressing GFP-tagged YAP can exhibit colloidal phase separation and formation of YAP liquid droplets in both nucleus and cytoplasm after treatment with 25% PEG to induce macromolecular crowding (*Cai et al., 2018*). Our findings suggest that endogenous YAP may also phase separate when it becomes stabilised and concentrated by clustering with Mask family proteins.

To further explore the possible physiological role of colloidal phase separation, we examined the subcellular localisation of Yki/YAP and Mask family proteins. Yki-GFP localises to intracellular granules in either the nucleus and cytoplasm of follicle cells or cultured *Drosophila* S2 cells (*Figure 7A*). Similarly, antibody staining for Yki and Mask reveals co-localisation in the same intracellular granules in *Drosophila* S2 cells (*Figure 7B*). In human cells in culture, immunostaining for YAP and Mask1 reveals a similarly granular pattern in both cytoplasm and nucleus (*Figure 7C*). These findings

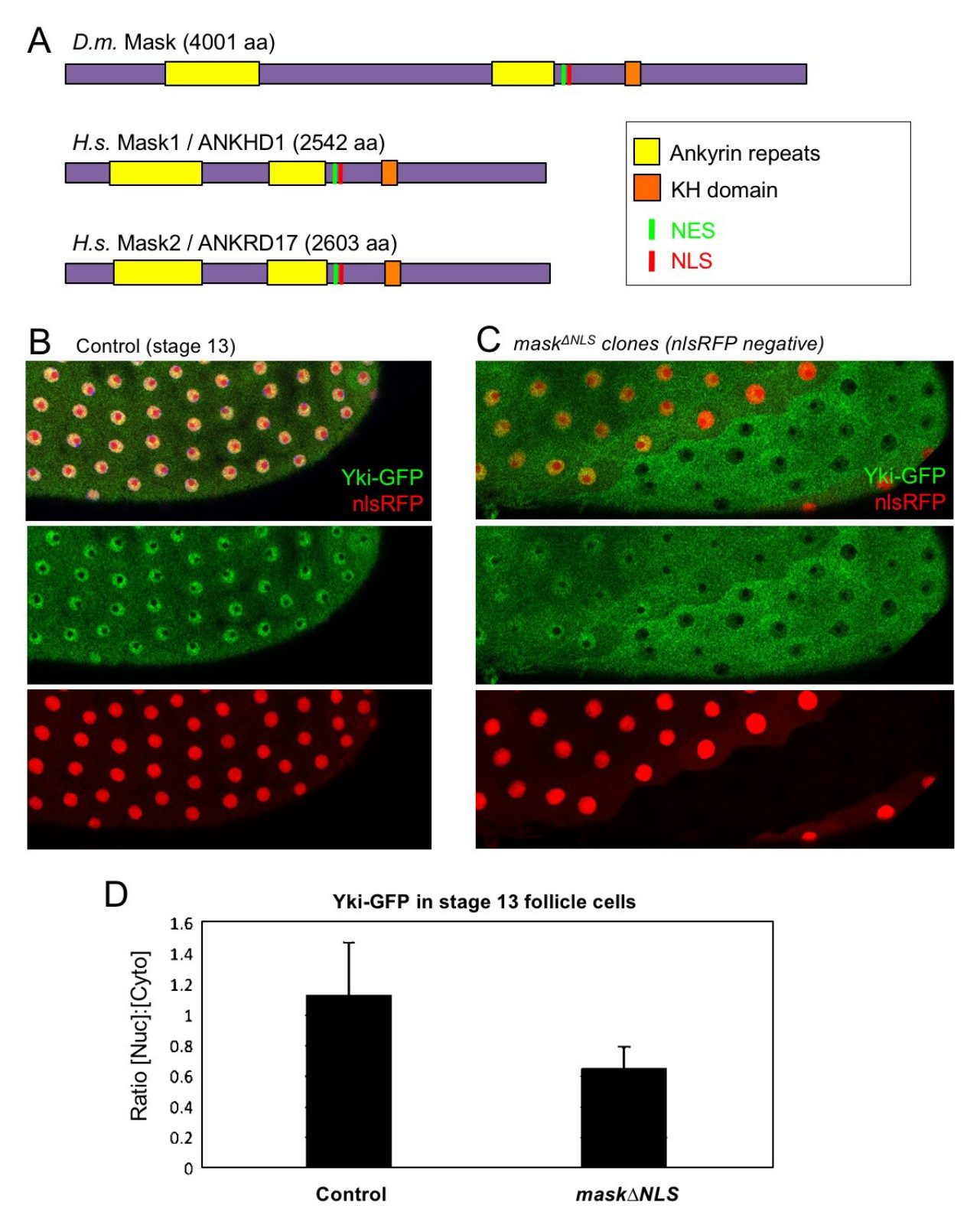

**Figure 2.** CRISPR-knockout of the Mask nuclear localisation signal leads to cytoplasmic accumulation of Yki. (**A**) Schematic diagram of the Mask family proteins and their conserved domains and motifs. (**B**) Stage 13 *Drosophila* egg chamber with endogenously tagged Yki-GFP (green) localised to the nucleus of flattening columnar cells. Nuclear localised RFP is also shown (red). (**C**) Stage 13 *Drosophila* egg chamber with endogenously tagged Yki-GFP (green) containing homozygous mutant clones for a CRISPR-knockin *maskΔNLS* mutant allele, marked by the absence of nuclear RFP (red). Note

*Figure 2 continued on next page*

*Figure 2 continued*

that Yki-GFP is no longer localised to the nucleus of flattening columnar cells in the *maskΔNLS* mutant clone. (D) Quantification of nuclear:cytoplasmic ratio of Yki-GFP in multiple *maskΔNLS* mutant clones as shown in (C) n = 22 clones.

The online version of this article includes the following figure supplement(s) for figure 2:

**Figure supplement 1.** The MaskΔNLS protein is expressed at normal levels and is localised to the cytoplasm.

indicate that endogenous Yki/YAP and Mask proteins co-localise and may cluster/polymerise to form intracellular granules (or 'hubs') as they interact with cytoplasmic 14-3-3 or nuclear TEAD (*Figure 7D*).

We previously showed that human Mask1/2 proteins can relocalise with YAP from the cytoplasm to the nucleus of cultured human cells in a density-dependent fashion (*Sidor et al., 2013*). We later found that integrin signalling via Src family kinases is the primary mechanism of YAP mechano-regulation in these cells and is crucial to regulate wound healing (*Elbediwy et al., 2018*; *Elbediwy et al., 2016*). We therefore tested whether the density-dependent subcellular localisation of Mask1/2 proteins was affected by loss of integrin adhesions (by removing Calcium), by loss of mechanical forces (by treatment with the actin drug Latrunculin), by loss of Src family kinase activity (by treatment with the Src inhibitor Dasatinib), or after scratch wounding (*Figure 8A,B*). In all cases, alterations in the localisation of Mask1/2 and YAP before and after treatment were the same, suggesting that binding of Mask1/2 to YAP determines its localisation (8A-C). Finally, we previously identified a physiological role for YAP (and TAZ) in skin wound healing, which causes YAP to localise to the nucleus in leading edge skin epithelial cells (*Elbediwy et al., 2016*). Mask1/2 are similarly nuclear localised during skin wound healing (*Figure 8D*), supporting the notion that these proteins are co-regulated in response to both mechanical cues and tissue damage in vitro and in vivo.

## Discussion

Our results shed light on the molecular mechanisms governing the activity of Yki/YAP family of transcriptional co-activators. Dynamic and continuous nucleo-cytoplasmic shutting of both Yki and YAP is well established (*Ege et al., 2018*; *Manning et al., 2018*). Dynamic shutting enables the subcellular localisation of the bulk of Yki/YAP protein to be governed simply by the availability of binding partners such as cytoplasmic 14-3-3 proteins or nuclear Sd/TEAD. Indeed, shuttling enables Hippo signalling to control bulk Yki/YAP subcellular localisation by regulating phosphorylation of Yki/YAP and thus its binding to 14-3-3 proteins. In the absence of a Hippo signal, unphosphorylated Yki/YAP binds primarily to nuclear Sd/TEAD and thus adopts a bulk nuclear localisation while maintaining shuttling to allow continuous sensing of Hippo activity in the cytoplasm. Our results reveal a novel molecular mechanism for Yki/YAP nuclear import by Mask family proteins.

We have established the essential requirement for the canonical nuclear localisation signal (NLS) in the *Drosophila* Mask protein, which is conserved in mammalian Mask1 (ANKHD1) and Mask2 (ANKRD17). Deletion of the Mask NLS by CRISPR knockin caused accumulation of Yki-GFP within mutant clones of cells. Furthermore, attachment of an ectopic NLS to Yki was sufficient to bypass the requirement for Mask in Yki-dependent cell proliferation and tissue growth. Finally, our results demonstrated a conserved requirement for human Mask1 in driving YAP to the nucleus in human cells.

Although the above findings demonstrate the primacy of the Mask NLS in mediating nuclear import, it is possible that non-canonical 'RaDAR' nuclear import motifs that bind RanGDP within the ankyrin repeat domains of Mask family proteins may also contribute, although the corresponding residues in Mask proteins are not strongly hydrophobic so may only function weakly, if at all (*Lu et al., 2014*). Furthermore, it is possible that Importin-alpha binding non-canonical nuclear import motifs within Yki (*Wang et al., 2016*) and a Ran-independent nuclear import sequence in YAP/TAZ (*Kofler et al., 2018*) may also contribute to import. However, these non-canonical motifs are not sufficient to maintain normal nuclear localisation of either Yki or YAP in the absence of Mask proteins.

In order for Mask to promote nuclear-cytoplasmic shuttling of Yki, the Mask-Yki interaction must be weak and constitutive – so that a pool of Yki protein is always dynamically shuttling. The

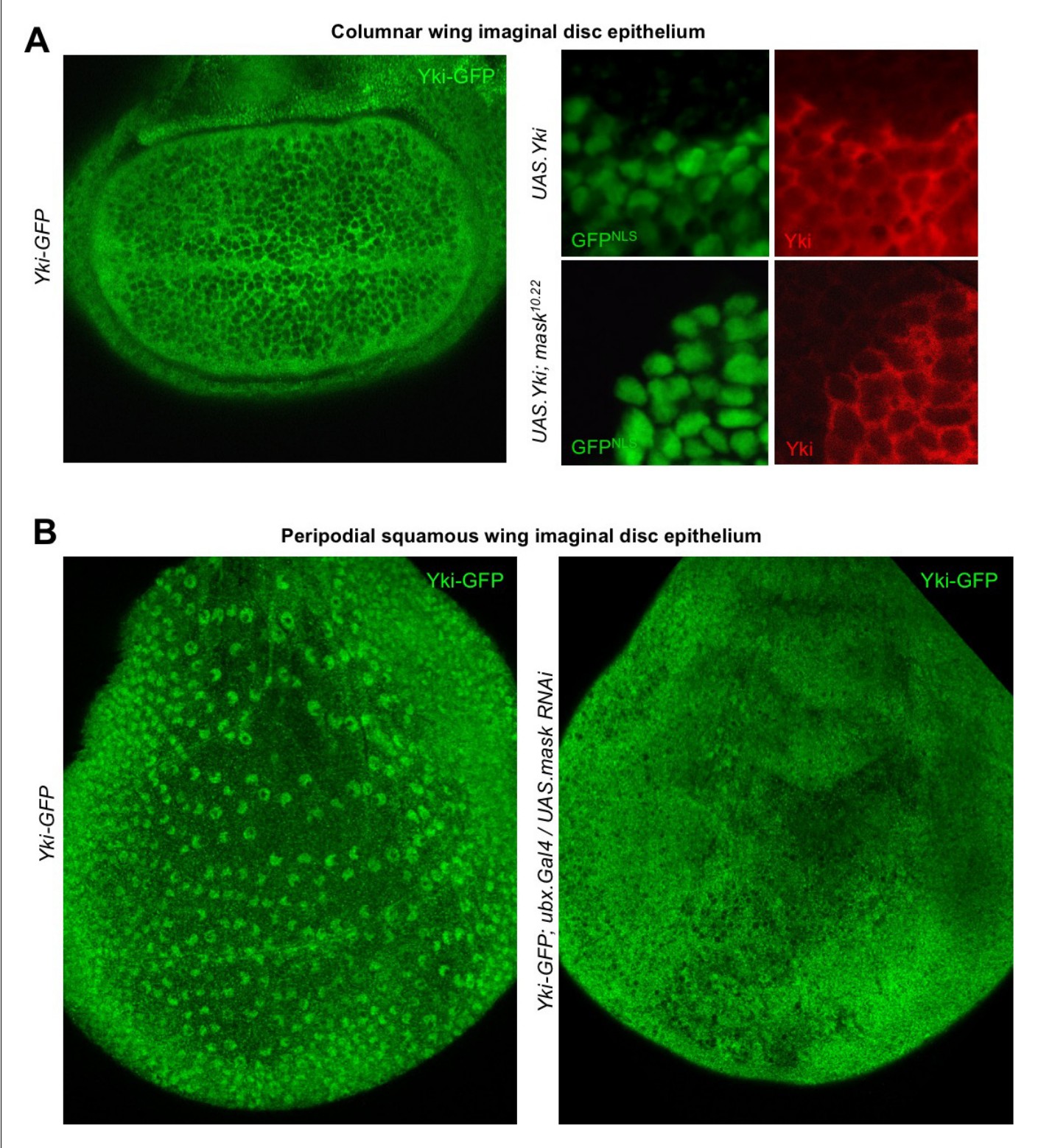

**Figure 3.** Mask is required to promote nuclear localisation of Yki in developing *Drosophila* wings. (**A**) Columnar epithelium of the wing imaginal disc from third instar larva showing mostly cytoplasmic localisation of endogenously tagged Yki-GFP (left). MARCM clones expressing Yki with the GAL4/ UAS system show both nuclear and cytoplasmic Yki localisation, as detected by Yki antibody staining (nlsGFP+). MARCM clones for a null *mask* mutant allele show reduced nuclear Yki (nlsGFP+). n = 6 clones. (**B**) Squamous peripodial epithelium from the wing imaginal disc from third instar larva showing

*Figure 3 continued on next page*

*Figure 3 continued*

mostly nuclear localisation of endogenously tagged Yki-GFP (left). Silencing of Mask expression by *ubx.Gal4* driven *UAS.mask-IR* (inverted repeat hairpin RNAi) in the peripodial epithelium is sufficient to reduce nuclear Yki-GFP localisation.

regulation of the bulk distribution of Yki is then determined simply by its choice of strong binding partner, namely either 14-3-3 in the cytoplasm or Sd in the nucleus, as mentioned above. Thus, there is always a 14-3-3 bound pool of Yki, a Sd bound pool of Yki and a third pool of Yki that can shuttle between the cytoplasm and nucleus by binding to Mask proteins and possibly other import/export factors. The relative proportions of the 14-3-3 bound and Sd bound pools determines the bulk distribution of Yki. Hippo signalling affects the 14-3-3 interaction (by Yki phosphorylation), while expression levels of the *sd* gene affects the Sd interaction. According to this model, there is no requirement for regulation of the Yki-Mask interaction and we have no evidence for it.

Our findings also establish a novel role for mammalian Mask1/2 in stabilising YAP protein levels. Unlike *Drosophila* Yki, YAP/TAZ stability is regulated by two SCF-type E3 ubiquitin ligases, beta-TrCP and FBXW7, which earmark YAP/TAZ for degradation after phosphorylation by LATS and CK1 on a specific phosphodegron containing Ser381 (*Tu et al., 2014*; *Zhang et al., 2016*; *Zhao et al., 2010*). Consequently, while loss of *Drosophila* Mask does not reduce total Yki levels, loss of Mask1/2 causes a dramatic loss of YAP protein, suggesting that binding of Mask1/2 to YAP is essential to prevent YAP degradation.

An interesting consequence of the role of Mask1/2 in stabilising YAP protein is that overexpression of Mask proteins is sufficient to dramatically raise the concentration of YAP within the cytoplasm, even causing formation of what appear to be YAP liquid droplets in human cells. Similar YAP liquid droplets have recently been shown to occur by phase separation after treatment of cells expressing GFP-tagged YAP with 25% PEG to drive macromolecular crowding (*Cai et al., 2018*). Whether phase-separation of YAP has a physiological role is still unknown, but the possibility is supported by our finding that endogenous YAP can also be observed to form droplets when in a complex with Mask1 at high concentrations. Interestingly, phase-separation has been proposed to support formation of transcriptional enhancer complexes ('transcription factories') in the nucleus (*Boehning et al., 2018*; *Boija et al., 2018*; *Hnisz et al., 2017*; *Iborra, 2007*; *Jackson et al., 1993*; *Lu et al., 2018*; *Sabari et al., 2018*), and we speculate that Mask proteins might function in this manner to promote transcription. Furthermore, Mask family proteins contain a conserved RNA binding KH-domain, and several RNA binding proteins have key roles in colloidal phase separation (*Anderson and Kedersha, 2006*; *Lin et al., 2015*; *Maharana et al., 2018*; *Mao et al., 2011*; *Molliex et al., 2015*; *Ramaswami et al., 2013*; *Weber and Brangwynne, 2012*). Finally, the fact that Mask family proteins contain two YAP-binding ankyrin repeat domains provides a plausible multivalent interaction mechanism by which Mask proteins could promote clustering of multiple YAP molecules in either the nucleus or cytoplasm, although further work is needed to explore this possibility (*Figure 7D*).

In conclusion, our results identify novel molecular mechanisms regulating the Yki/YAP/TAZ family of transcriptional co-activators. Mask family proteins promote nuclear import of both Yki and YAP, while also acting to stabilise YAP protein levels, which can lead to what appear to be YAP droplets at high concentrations. Endogenous Mask1/2 and YAP co-localise in intracellular granules in both the nucleus and cytoplasm, and the nucleo-cytoplasmic distribution of Mask1/2 and YAP is co-regulated in response to mechanical cues in human cells, most likely through direct association of the two proteins. These functions of Mask family proteins are crucial to enable dynamic nuclear-cytoplasmic shuttling and thus regulation of these co-activators by Hippo signalling and Integrin signalling to control cellular behaviour.

## Materials and methods

### *Drosophila* genetics

All *Drosophila* strains have been previously described in *Sidor et al. (2013)* and *Fletcher et al. (2018)* or are available from Bloomington Drosophila Stock Centre, with the exception of the *mask*$^{\Delta NLS}$ allele, which was generated in this study, as follows. CRISPR/Cas9-mediated genome

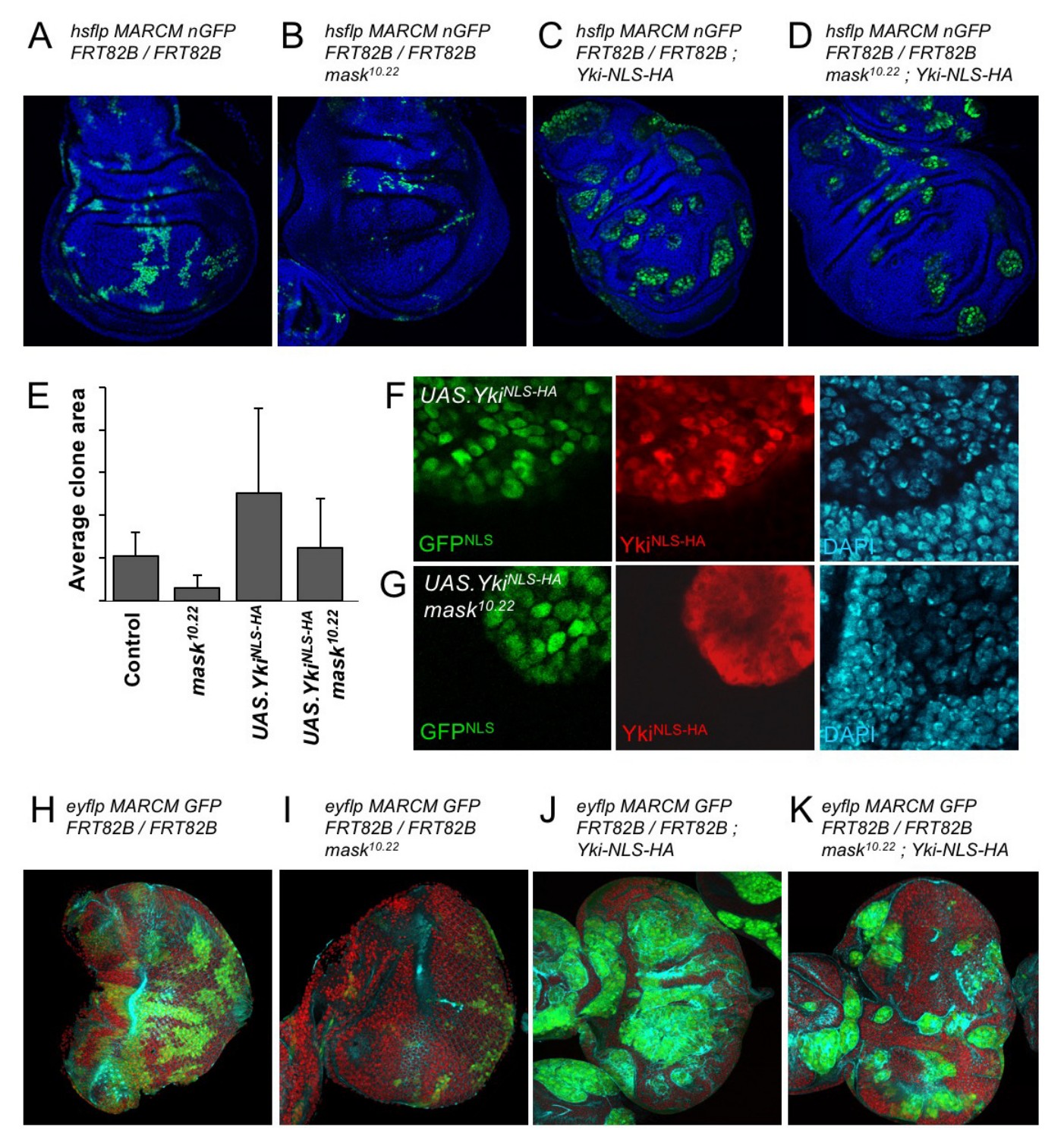

**Figure 4.** Addition of an NLS to Yki enables it to enter the nucleus and drive clonal growth in *mask* mutant cells. (A) Wing imaginal disc containing control MARCM clones (nlsGFP+, green). (B) Wing imaginal disc containing null mutant *mask* MARCM clones (nlsGFP+, green) are almost eliminated. (C) Wing imaginal disc containing control MARCM clones also expressing nlsYki[HA] (nlsGFP+, green) proliferate more strongly than controls. (D) Wing imaginal disc containing null mutant *mask* MARCM clones also expressing nlsYki[HA] (nlsGFP+, green) are rescued for their growth and survival to a level similar to wild-type clones. (E) Quantification of average clone sizes in A-D. (F) Close-up view of control MARCM clones expressing nlsYki[HA] in red (marked by nlsGFP+, green). DAPI in blue. (G) Close-up view of null mutant *mask* MARCM clones expressing nlsYki[HA] in red (marked by nlsGFP+, green). DAPI in blue. (H) Eye imaginal disc containing control MARCM clones (nlsGFP+, green). (I) Eye imaginal disc containing null mutant *mask*

*Figure 4 continued on next page*

Figure 4 continued

MARCM clones (nlsGFP+, green) are almost eliminated. (J) Eye imaginal disc containing control MARCM clones also expressing nlsYki[HA] (nlsGFP+, green) proliferate more strongly than controls. (K) Eye imaginal disc containing null mutant *mask* MARCM clones also expressing nlsYki[HA] (nlsGFP+, green) are rescued for their growth and survival to a level similar to wild-type clones.

editing by homology-dependent repair (HDR) of *Mask/CG33106* 3R (95F3-95F5) using one guide RNA and a dsDNA plasmid donor was used to knockin a cassette, PBacDsRed, into the injection strain *w1118.* Genome editing involved deleting a 48 bp nuclear localization sequence (NLS) 'CGTCGTCGTGAGCGCaagaagaagaagaagatggagaagaaagaggagaagCGCCGTCAA' and replacement by PBacDsRed. This cassette contains a 3XP3-DsRed that facilitated the genetic screening and was then excised by Piggy Bac transposase to enable normal expression of the gene. Only one TTAA motif was left after transposition that is embedded in nearby mutated silenced mutation from GTCAAG to GTtAAG encoded in VK. The following guide RNAs were used:

### gRNA 1

CRISPR Target Site [PAM]: TCTTCTTGCGCTCACGACGA[CGG]
CRISPR Target Strand in the Genome: minus
Cutting Site: −4031 nt from stop codon of *Mask*
Distance: +87 nt from upstream breakpoint to cutting site of Cas9
−55 nt from downstream breakpoint to cutting site of Cas9
Off target: 0; GC%: 55%; T number at 17th to 20th nt: 0;
Guide RNA Primers:
Sense oligo 5'-CTTCGTCTTCTTGCGCTCACGACGA
Antisense oligo 5'-AAACTCGTCGTGAGCGCAAGAAGAC
PAM mutation: not required
Upstream Homology Arm: 1011 bp, −5128 nt to −4118 nt from stop codon of *Mask*
Downstream Homology Arm: 1001 bp, −3975 nt to −2975 nt from stop codon of *Mask*

### gRNA 2

CRISPR Target Site [PAM]: CTTGCGCTCACGACGACGGG[CGG]
CRISPR Target Strand in the Genome: minus
Cutting Site: −4034 nt from stop codon of *Mask*
Distance: +83 nt from upstream breakpoint to cutting site of Cas9
−59 nt from downstream breakpoint to cutting site of Cas9
Off target: 1 on 2L:20826776..20826798
GC%: 70%; T number at 17th to 20th nt: 0;
Guide RNA Primers:
Sense oligo 5'-CTTCGCTTGCGCTCACGACGACGGG
Antisense oligo 5'-AAACCCCGTCGTCGTGAGCGCAAGC
PAM mutation: not required
Upstream Homology Arm: same as above
Downstream Homology Arm: same as above

Validation of the exised allele was performed by sequencing after PCR across the excised region. Genomic DNA was obtained from single fly of each stock following single-fly DNA prep.Injection strain *w1118*was used as a negative control. PCR was performed using KOD-FX (TOYOBO) on Bio-Rad S1000 Thermal Cycler.100bp DNA Ladder from GenePurewas used as reference. There was a 737bp-product in the injection strain control. The product length was 2373 bp before excision of the PBacDsRed cassette and 677 bp after excision using the following PCR primers:

OWG66235'-TCACCTGAGTGTGGTTGAGC
OWG66245'-GGACGGAGTTCGATGTGATT
The resulting excised PCR bands were sequenced following gel extraction and translated into protein sequence as follows:
WT Mask NLS containing region:

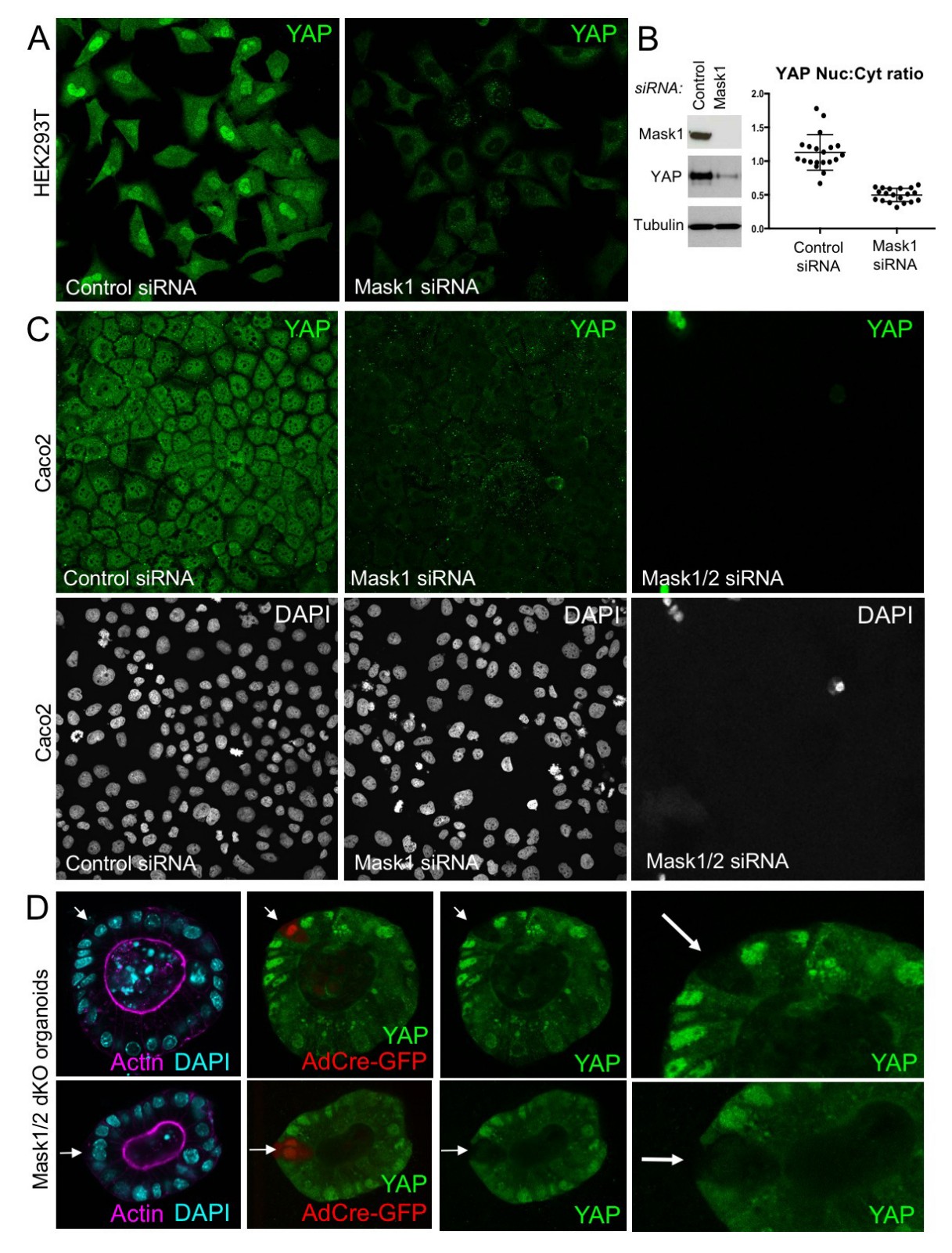

**Figure 5.** Mammalian Mask1/2 are required for nuclear localisation and stability of YAP. (**A**) HEK293T cells display nuclear YAP localisation after transfection with control scrambled siRNA, and low levels of cytoplasmic YAP localisation after transfection with Mask1 siRNA oligonucleotides. (**B**) Western analysis of Mask1 and YAP after transfection of HEK293T cells with control scrambled or Mask1 siRNA oligonucleotides (left). Quantification of the nuclear:cytoplasmic ratio of YAP subcellular localisation after transfection of HEK293T cells with control scrambled or Mask1 siRNA oligonucleotides

*Figure 5 continued on next page*

*Figure 5 continued*

as shown in A (right). (C) Caco2 intestinal epithelial cells transfected with control or Mask1 or Mask1/2 siRNA oligonucleotides and immunostained for YAP or stained with DAPI to mark nuclei. (D) Intestinal organoids freshly cultured from Mask1/2 double conditional knockout mice, in which Ad-Cre was used to induce clonal knockout of both Mask1 and Mask2 in single cells (marked by Ad-Cre-GFP in red). Note the dramatic reduction in YAP protein levels (n = 12 organoids).

The online version of this article includes the following figure supplement(s) for figure 5:

**Figure supplement 1.** Mask1 and two double siRNA transfected cells undergo cell death.

**Figure supplement 2.** Mask1 and 2 dKO cells induced by Adenoviral-Cre infection tend to be extruded from intestinal organoids.

**Figure supplement 3.** Mask1 regulates TAZ similarly to YAP in Caco2 cells.

240     KANKNASILLEELDLERTREESRKAAAAARRRERKKKKKMEKKEEKRR     QQQGNGPGGDD
MQGDDDDASDKDDDSDKDDEDEEAAPAAAREEGDSGIDQGSC
Mutant CRISPR knockin MaskΔNLS:
240    KANKNASILLEELDLERTREESRKAAAAQQQGNGPGGDDMQGDDDD    ASDKDDDSDKDDE-
DEEAAPAAAREEGDSGIDQGSC

## Induction of clones in *Drosophila*

Mosaic tissues were generated using the FLP/FRT and the MARCM system with a heat shock promoter (hs) to drive the expression of the FLP recombinase. Clones in imaginal discs were induced by heat shocking larvae at 60 hr (±12 hr) of development and larvae were dissected at the third instar stage. Clones in ovarian follicle cells were induced by heat shocking adult females fed with yeast for 3 days before dissection at various times after heat shock.

## Fluorescent immunostaining

### *Drosophila*

Ovaries and imaginal discs were dissected in PBS, fixed for 20 mins in 4% paraformaldehyde in PBS, washed for 30 min in PBS/0.1% Triton X-100 (PBST) and blocked for 15 min in 5% normal goat serum/PBST (PBST/NGS). Primary antibodies were diluted in PBST/NGS and samples were incubated overnight at 4°C (*Fletcher et al., 2018*). The following primary antibodies were used:

FITC-conjugated anti-GFP (1:400, Abcam)
Rabbit Anti-Mask (Michael Simon) 1:500 rat anti-Yki (Nic Tapon) 1:500
Mouse anti-V5 1:100 (Abcam)

Secondary antibodies (all from Molecular Probes, Invitrogen) were used at 1:500 for 2–4 hr prior to multiple washes in PBST and staining with DAPI at 1 µg/ml for 10-30mins prior to mounting on slides in Vectashield (Vector labs).

### Human cells

All cell lines were obtained from the Francis Crick Institute Cell Services Core Facility and are certified mycoplasma-free. Cells were fixed and stained by standard procedures using the following antibodies:

Goat anti-Mask1/ANKHD1 (Santa-Cruz) 1:100
Mouse anti-Mask2/ANKRD17 (Sigma) 1:100
Goat anti-Mask2/ANKRD17 (Santa-Cruz) 1:100
Rabbit anti-YAP (Santa-Cruz) 1:200
Alexa-fluor secondary antibodies (Invitrogen) 1:500

DNA was stained with DAPI. Samples were imaged with a Leica SP5 Confocal Microscope.

### siRNA methods

### siRNAs

three different pools of siRNAs were used at a final concentration of 300 nM (75 nM of each individual siRNA):

ON-TARGETplus Set of 4 Mask1/ANKHD1 (Dharmacon)

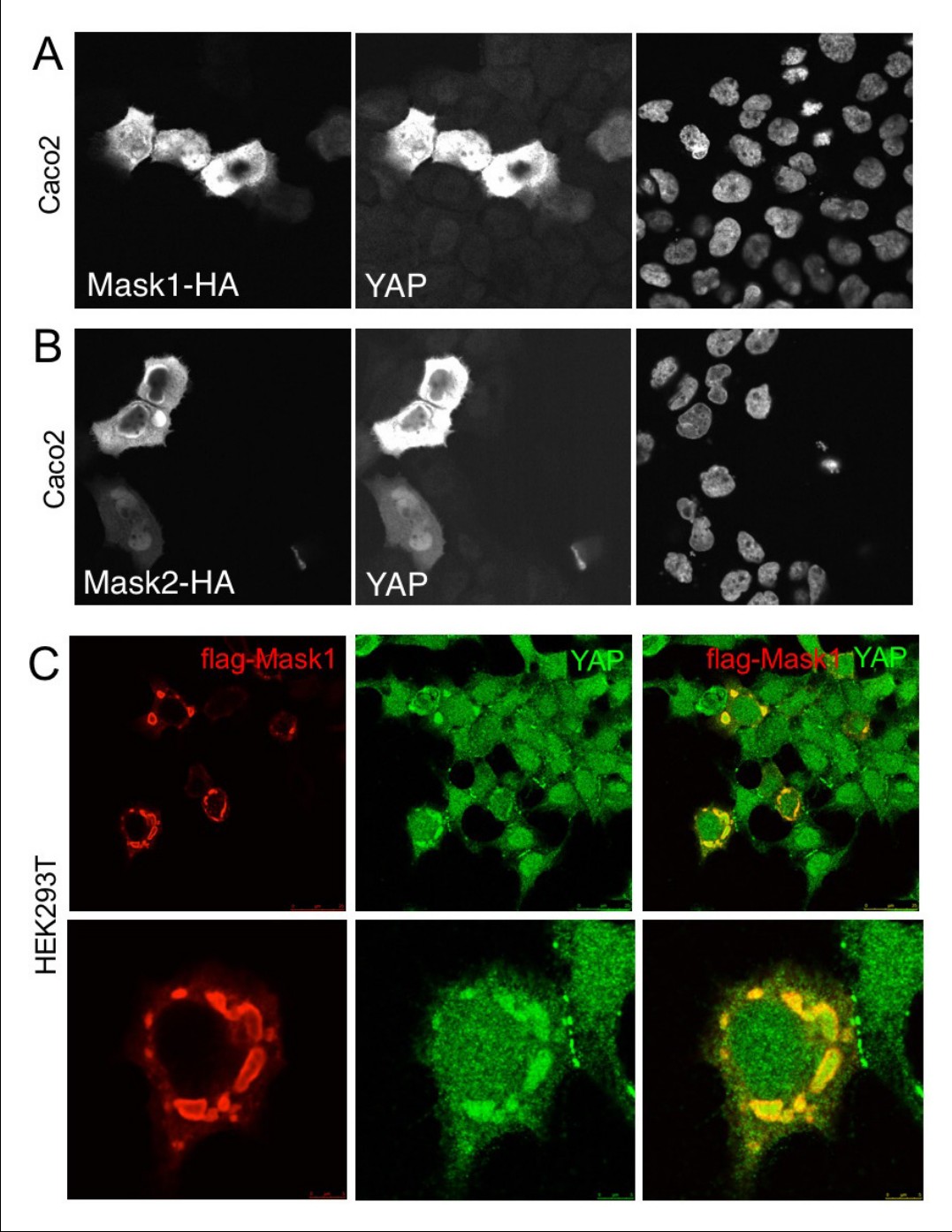

**Figure 6.** Overexpression of Mask1/2 stabilizes YAP and can induce phase separation of YAP. (**A**) Caco2 intestinal epithelial cells overexpressing HÁ-tagged Mask1, which induces stabilisation of endogenous YAP. DAPI marks nuclei in the right-hand panel. (**B**) Caco2 intestinal epithelial cells overexpressing HÁ-tagged Mask2, which also induces stabilisation of endogenous YAP. DAPI marks nuclei in the right-hand panel. (**C**) HEK293T cells overexpressing HÁ-tagged Mask1, which induces stabilisation of endogenous YAP and formation of large liquid-like droplets with smooth boundaries containing both proteins – suggestive of colloidal phase separation.

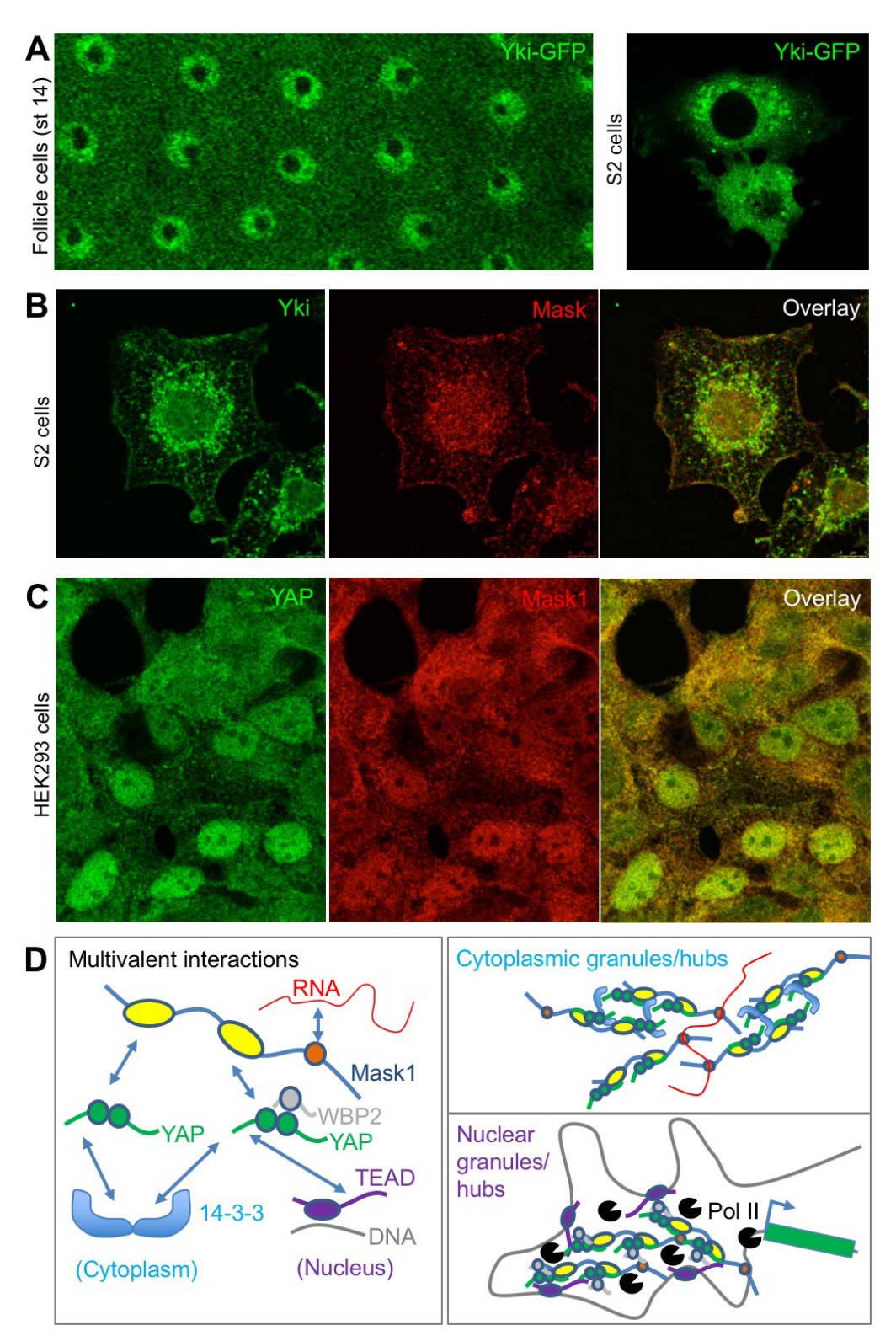

**Figure 7.** Endogenous Yki/YAP and Mask co-localise in intracellular granules in both nucleus and cytoplasm. (**A**) Yki-GFP exhibits a granular appearance in both nucleus and cytoplasm of flattened *Drosophila* follicle cells. Yki-GFP also exhibits a granular appearance in the cytoplasm of S2 cells. (**B**) Co-localisation of endogenous Yki and Mask in nuclear and cytoplasmic granules of S2 cells. (**C**) Co-localisation of endogenous YAP and Mask1 in nuclear

*Figure 7 continued on next page*

and cytoplasmic granules of HEK293 cells. (D) Schematic diagram of multivalent interactions between YAP binding partners and the possible formation of clusters (hubs/granules) in either the cytoplasm or nucleus.

ON-TARGETplus Set of 4 Mask2/ANKRD17(Dharmacon)
ON-TARGETplus Non-targeting scrambled pool (Dharmacon)

Cells were plated in 12 well plates and transfected with siRNA pools using the Lipofectamine RNAiMax reagent (invitrogen). Cells were harvested after 48 hr and analysed by immunostaining or western blot.

## Luciferase assay in S2 cells

The luciferase assay was performed in sextuplicates in a 96 well plate. Cells were transfected with 20 ng of each DNA plasmid per well. 48 hr after transfection, cells were lysed and tested for Luciferase and Renilla activity using the Dual-luciferase reporter assay system kit (Promega) and a luminometer.

## Mouse genetics

All experiments were carried out in accordance with the United Kingdom Animal Scientific Procedures Act (1986) and UK home office regulations under project license number 70/7926. For the generation of the Mask1/2 double conditional knockouts, the following EUCOMM ES cell lines were used:

Ankhd1$^{tm1a(KOMP)Wtsi}$ EPD0694_2_A02
Ankhd1$^{tm1a(KOMP)Wtsi}$ EPD0694_2_D03
Ankrd17$^{tm1a(EUCOMM)Hmgu}$ HEPD0725_3_D01
Ankrd17$^{tm1a(EUCOMM)Hmgu}$ HEPD0725_3_B04

## Organoid cultures

Intestinal crypts were isolated from the proximal part of the small intestine of 6–10 weeks Mask1 fl/fl Mask2 fl/fl mice (generated from EUCOMM ES cells for this study) as follows. The whole gut was harvested and washed in cold DPBS (#14190250, Gibco). The most proximal 5 cm were cut open, and the villi scraped off with a coverslip. The remaining tissue was cut in 0.5 cm pieces, washed several times by pipetting up and down in 8 mL fresh DPBS, and incubated in DPBS + 2 mM EDTA for 30 min at 4°C. Crypts were then mechanically extracted by vigorous shaking in DPBS, and filtered through a 70 µM Nylon cell strainer (#352350, Falcon). After several low speed washes in ADF-12 (#12634–010, Gibco), isolated crypts were resuspended and plated in Matrigel (#354230, Corning). Organoids were cultured in IntestiCult medium (#06005, Stemcell technologies) supplemented with Primocin antibiotic (1:50, ant-pm-2, Invivogen), either in 24-well plates for maintenance, or in 8-well chambers (#80827, Ibidi) for Ad-Cre-GFP infection and immunostaining. Passages were performed by resuspending Matrigel-embedded organoids in cold DPBS and transferring them to a Falcon tube using a 2 mL syringe with a 27 G ½ needle (BD Microlance #300635) to break them up. After two low speed washes in ADF-12, organoids were resuspended and plated in Matrigel.

## Organoid infection with Ad-Cre-GFP

Cre recombinase was expressed in organoids using Recombinant Adenovirus Ad-Cre-GFP from SignaGen Laboratories (#SL100706), which co-expresses GFP as a marker. The Ad-Cre-GFP virus was diluted in IntestiCult Medium to a concentration of $1.10^6$ pfu/µl. Organoid infection was performed using the Mix and Seed approach from 5-day old Mask1 fl/fl; Mask2 fl/fl organoids were passaged into fresh Matrigel. Ad-Cre-GFP was added to the unpolymerized Matrigel-organoid mix at a final concentration of $5.10^6$ pfu per 100 µl. Organoids were then plated in 8-well chambers and grown in IntestiCult medium for 3 days at 37°C.

## Organoid fixation and staining

Organoids embedded in Matrigel were washed once in PBS, then incubated in fix solution (PBS + 4% formaldehyde) for 30 min. After two washes (5 min) in PBS, they were incubated in quenching

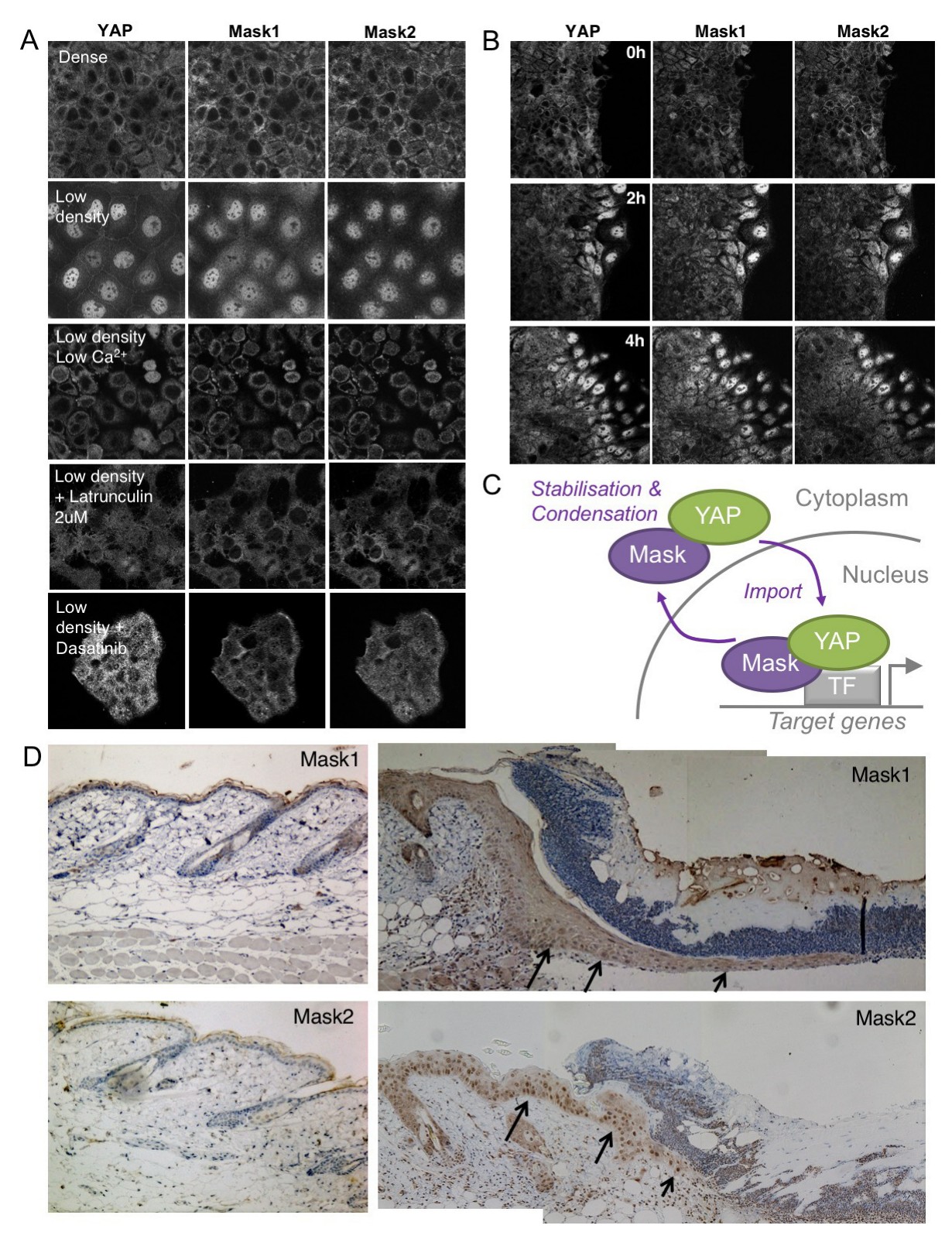

**Figure 8.** Co-regulation of YAP and Mask1/2 both in vitro and in vivo. (A) Caco2 intestinal epithelial cells immunostained for YAP, Mask1 and Mask2 at different densities and following treatments to interrupt Integrin-Src mediated mechanotransduction (low Calcium, Latrunculin, or Dasatinib). (B) Caco2 intestinal epithelial cells immunostained for YAP, Mask1 and Mask2 at various times after scratch wounding of the epithelium. (C) Schematic diagram of

*Figure 8 continued on next page*

*Figure 8 continued*

Mask-YAP interactions in the nucleus and cytoplasm and their co-regulation during nucleo-cytoplasmic shuttling. (**D**) Mouse skin epithelium before (left) and after (right) wounding, showing upregulation and nuclear localisation of Mask1/2 in vivo.

solution (PBS + 50 mM NH4Cl), then washed again 3 times 5 min in PBS. Organoids were permeabilised in PBST (PBS + 0.3% Triton) for 30 min, then blocked for 30 min in Blocking solution (PBS + 0.1% Triton + 5% NGS). Immunostaining was performed by incubating with Mouse Anti-YAP 63.7 from Santa Cruz (sc-101199; 1 in 100) in blocking solution overnight at 4°C in a wet chamber. After three 30 min washes in PBS, organoids were incubated in secondary antibody, DAPI and Phalloidin-647 for 1 hr and 30 min at room temperature and washed again 3 times. After removal of the last wash, 80% glycerol was added to the wells at least one hour prior to imaging. Imaging was performed with a Leica SP5 laser-scanning confocal microscope.

## Acknowledgements

We thank Hannah Vanyai for maintaining the mouse colony and breeding the Mask1/2 double floxed animals.

## Additional information

### Funding

| Funder | Grant reference number | Author |
| --- | --- | --- |
| Wellcome | FC001180 | Barry J Thompson |
| Cancer Research UK | FC001180 | Barry J Thompson |
| Medical Research Council | FC001180 | Barry J Thompson |

The funders had no role in study design, data collection and interpretation, or the decision to submit the work for publication.

### Author contributions

Clara Sidor, Conceptualization, Data curation, Formal analysis, Investigation, Methodology; Nerea Borreguero-Munoz, Oriane Guillermin, Investigation, Methodology; Georgina C Fletcher, Supervision, Investigation, Methodology; Ahmed Elbediwy, Investigation; Barry J Thompson, Conceptualization, Supervision, Funding acquisition, Writing—original draft, Project administration, Writing—review and editing

### Author ORCIDs

Ahmed Elbediwy (iD) http://orcid.org/0000-0002-2102-7339
Barry J Thompson (iD) https://orcid.org/0000-0002-0103-040X

### Ethics

All experiments were carried out in accordance with the United Kingdom 462 Animal Scientific Procedures Act (1986) and UK home office regulations 463 under project license number 70/7926.

### Decision letter and Author response

Decision letter https://doi.org/10.7554/eLife.48601.SA1
Author response https://doi.org/10.7554/eLife.48601.SA2

## Additional files

### Supplementary files

• Source data 1. Primary data file.This .xlsx file contains the primary data used to generate the graphs shown in the main figures.

• Transparent reporting form

### Data availability

All data generated or analysed during this study are included in the manuscript and supporting files.

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
