## [Decision Letter]

**Acceptance summary:**

Mask family proteins have been implicated as positive regulators of the Hippo signaling pathway. However, the mechanistic basis for Mask family protein regulation has not been established. The study by Sidor et al. demonstrates that Mask family proteins bind to the transcriptional regulator Yorkie and promote nuclear import. This finding represents an important advance in understanding the function of the Hippo pathway.

**Decision letter after peer review:**

Thank you for submitting your article "Mask family proteins ANKHD1 and ANKRD17 regulate YAP nuclear import, stability and phase separation" for consideration by *eLife*. Your article has been reviewed by two peer reviewers, one of whom is a member of our Board of Reviewing Editors, and the evaluation has been overseen by Kevin Struhl as the Senior Editor. The following individual involved in the review of your submission has agreed to reveal their identity: Junhao Mao (Reviewer #2).

The reviewers have discussed the reviews with one another and the Reviewing Editor has drafted this decision to help you prepare a revised submission.

Summary:

This is an interesting study that was designed to re-examine the role of Mask proteins in the Hippo pathway. Previous analysis by these authors demonstrated that Mask is required for Yki signaling, that it binds Yki, and that it may act as a co-activator (and not nucleocytoplasmic transport). Here the authors employ YFP-Yki (rather than immunostaining) and demonstrate that Mask is required for Yki transport to the nucleus. The data presented concerning a role for Mask in Yki nuclear import are convincing. However, there are a number of issues that remain to be addressed.

Essential revisions:

1) The Mask∆NLS allele studies are important. However, this allele is not described. The Materials and methods section refers to previous literature, but this information is not obvious. The authors need to state how the allele was made and the nature of the mutation. The Mask∆NLS phenotype resembles a Mask loss of function allele – is the Mask∆NLS protein expressed at WT levels? What is the localization of the Mask∆NLS protein?

2) Yki nuclear accumulation is regulated. No information is provided concerning how Mask contributes to Yki nuclear accumulation. For example, Yki binds Mask – is this interaction regulated or is it constitutive? Some insight into the mechanism of regulated nuclear accumulation of Yki is needed – what is the role of Mask? For example, is there a role for Yki phosphorylation in Mask function? etc.

3) Mask regulation of YAP nuclear import in mammalian cells is not fully documented. The authors show co-localization of Mask and Yki/YAP in vitro and in vivo (Fig7/8); however, the interpretation of siRNA experiments in 293 cells (Figure 5) is complicated by dramatically decreased YAP expression. This study can be strengthened by measuring its relative levels in nuclear and cytoplasmic fractions by immunoblots. The Mask1/2 DKO organoid assay (Figure 5D) has a similar concern, as only one or two cells carry Mask KO and have much lower YAP levels.

4) Do Mask1/2 also regulate TAZ nuclear localization or stability?

5) Figure 6. Conclusions concerning the role of Mask in YAP phase separation are speculative – this should be noted. This section of the manuscript represents a weakness of the overall study. The authors should test whether mutated forms of Mask, including ∆NLS and ∆NES affect YAP protein aggregation in different cellular compartments.

---

## [Author Response]

Essential revisions:1) The Mask∆NLS allele studies are important. However, this allele is not described. The Materials and methods section refers to previous literature, but this information is not obvious. The authors need to state how the allele was made and the nature of the mutation. The Mask∆NLS phenotype resembles a Mask loss of function allele – is the Mask∆NLS protein expressed at WT levels? What is the localization of the Mask∆NLS protein?

As requested, we now provide a comprehensive description of the generation of the *mask∆NLS* allele in the revised Materials and methods section. We also provide new data to show that the mutant protein is expressed normally and localised to the cytoplasm (Figure 2—figure supplement 1).

2) Yki nuclear accumulation is regulated. No information is provided concerning how Mask contributes to Yki nuclear accumulation. For example, Yki binds Mask – is this interaction regulated or is it constitutive? Some insight into the mechanism of regulated nuclear accumulation of Yki is needed – what is the role of Mask? For example, is there a role for Yki phosphorylation in Mask function? etc.

To clarify, we provide the following new discussion to address these questions. “In order for Mask to promote nuclear-cytoplasmic shuttling of Yki, the Mask-Yki interaction must be weak and constitutive – so that a pool of Yki protein is always dynamically shuttling. The regulation of the bulk distribution of Yki is then determined simply by its choice of strong binding partner, namely either 14-3-3 in the cytoplasm or Sd in the nucleus. Thus, there is always a 14-3-3 bound pool of Yki, a Sd bound pool of Yki and a third pool of Yki that can shuttle between the cytoplasm and nucleus by binding to Mask proteins and possibly other import/export factors. The relative proportions of the 14-3-3 bound and Sd bound pools determines the bulk distribution of Yki. Hippo signalling affects the 14-3-3 interaction (by Yki phosphorylation), while expression levels of the *sd* gene affects the Sd interaction. According to this model, there is no requirement for regulation of the Yki-Mask interaction and we have no evidence for it.”

3) Mask regulation of YAP nuclear import in mammalian cells is not fully documented. The authors show co-localization of Mask and Yki/YAP in vitro and in vivo (Fig7/8); however, the interpretation of siRNA experiments in 293 cells (Figure 5) is complicated by dramatically decreased YAP expression. This study can be strengthened by measuring its relative levels in nuclear and cytoplasmic fractions by immunoblots. The Mask1/2 DKO organoid assay (Figure 5D) has a similar concern, as only one or two cells carry Mask KO and have much lower YAP levels.

We have quantified the level of YAP protein by immunoblot and also quantified the nuclear/cytoplasmic ratio of YAP upon Mask siRNA in Figure 5B. Importantly, the reduced level of YAP does not prevent us from measuring the nuclear/cytoplasmic ratio. We therefore feel the fractionation of cells and immunoblotting will not provide a better measure of YAP subcellular localisation than the immunostaining we have already performed in this experiment, particularly as fractionation often leads to contamination of nuclear and cytoplasmic pools. The organoid DKO indeed has such low levels of YAP that we cannot measure its distribution, but this is the only conclusion that we draw from this experiment: that Mask1/2 DKO strongly reduces YAP levels in organoids.

4) Do Mask1/2 also regulate TAZ nuclear localization or stability?

Yes, we see identical effects of Mask siRNA on both YAP and TAZ, as expected. The new TAZ data are provided in Figure 5—figure supplement 3.

5) Figure 6. Conclusions concerning the role of Mask in YAP phase separation are speculative – this should be noted. This section of the manuscript represents a weakness of the overall study. The authors should test whether mutated forms of Mask, including ∆NLS and ∆NES affect YAP protein aggregation in different cellular compartments.

We agree and have modified the language to emphasise the speculative nature of these conclusions, including removing the term ‘phase separation’ from the title and Abstract. As requested, we have attempted to test whether expression of Mask1∆NLS or ∆NES affects YAP protein aggregation, but both constructs accumulate strongly and aggregate in the cytoplasm due to overexpression.